# The Interaction of Positive and Negative Relationship Characteristics and Their Association with Relationship and Individual Health Outcomes in Older Couples

**DOI:** 10.3390/bs14111017

**Published:** 2024-11-01

**Authors:** Suzanne Bartle-Haring, Jie Hu, Lorraine Mion, Millie Ash

**Affiliations:** 1Department of Human Sciences, Couple and Family Therapy, The Ohio State University, Columbus, OH 43210, USA; ash.188@osu.edu; 2College of Nursing, The Ohio State University, Columbus, OH 43210, USA; hu.1348@osu.edu (J.H.); mion.3@osu.edu (L.M.)

**Keywords:** relationship satisfaction relationship quality, dyadic adjustment, older couples, dyadic analyses

## Abstract

The purpose of this study was to examine how positive and negative relationship characteristics and their interaction were predictive of global relationship happiness, psychological symptoms, and physical health in a large sample of older couples. The NSHAP Wave 2 partner data *n* = 955) were used to estimate both actor (within person) and partner (between person) effects using the Actor Partner Interdependence Model with Moderation. Global relationship happiness was predicted by the interaction of positive and negative characteristics, suggesting that more positive characteristics are only predictive of greater happiness in the presence of some negative characteristics. Male partners’ depressive symptoms were predicted by the female partners’ interaction of positive and negative characteristics, suggesting that negative characteristics were only associated with male partner depressive symptoms when positive characteristics were rated very low by their partner. Positive and negative characteristics were predictive of anxiety and stress but not their interaction, and only the male partner’s perception of positive characteristics was predictive of the female partner’s physical health. These results are discussed in the context of controversy over the measurement of relationship satisfaction in the field.

## 1. Introduction

In the fields of personal relationships, couple and family therapy, and family science, among others, terms like relationship satisfaction, dyadic adjustment, and relationship quality are used interchangeably. There are also a myriad of assessment instruments that use these terms but may be measuring essentially the same thing (i.e., Dyadic Adjustment Scale, Relationship Quality Inventory, Couple Satisfaction Inventory). Even with this, we continue to be challenged by how to conceptualize and measure relationship quality, satisfaction, or adjustment [1,2]. Whichever term one chooses to use, there seems to be at least two opposing views: either satisfaction, quality, or adjustment is in the “eye of the beholder”, that is, viewed as a global measure of the relationship; or there are certain characteristics (e.g., spending time together, low levels of conflict) that need to occur in relationships in order to be considered satisfying, of high quality, or well adjusted. The drawback with the latter viewpoint is that these characteristics are added together and formed into a single construct without considering how these characteristics might interact with each other [3]. For example, if a couple “spends time together”, that characteristic is counted toward good quality or adjustment, while if they have “high levels of conflict”, that characteristic is counted against adjustment. However, it is possible that these two characteristics interact in some way to influence the quality of the relationship such that spending time together protects against the negative impact from higher levels of conflict or that spending more time together increases the opportunity for more conflict [3].

Most of the literature would suggest that satisfying relationships promote both mental and physical health [4,5,6,7,8]. However, this research is conducted using instruments that fall within the two categories of measures discussed above, either as a global measure or as a measure that suggests certain characteristics that need to be present for relationships to be considered as high quality or satisfactory. Rarely do researchers consider more nuanced ways of assessing relationships and their associations with physical and mental health, and even more rarely do researchers investigate these among older couples.

Clarity in how we conceptualize and measure relationship quality has implications for what we may teach to those interested in improving their relationships or preventing relationships from “going south” and how we might intervene when relationships are distressing for all couples. In a recent review of assessments for marital quality [1], the Dyadic Adjustment Scale (DAS) [9] was found to be the most often used instrument, while most studies used single-dimension measures of relationship quality. Delatorre and Wagner [1] concluded that although there are many instruments available to assess relationship satisfaction, quality, and/or adjustment, they do not provide a clear definition of the constructs, and there is little consensus on the meaning of the terms adopted. There also appears to be a lack of theoretical foundation to most of the instruments, including DAS, which was developed empirically using items from other relationship measures. Thus, although there is a vast amount of literature about relationship satisfaction, adjustment, or quality, there is little consensus on their meanings. In reviewing titles and methods sections of three family science journals (*Journal of Marriage and Family, Journal of Family Psychology, Family Relations*) over the last 6 years, there were 45 papers that used relationship satisfaction, quality, or adjustment in the title. None of the papers utilized instruments related to the title constructs, and only three studies investigated the interaction of positive and negative characteristics in the relationship.

Without some understanding of the “balance” of positive and negative characteristics that allow for an overall satisfying (in the eye of the beholder) relationship, we will continue to develop relationship education and interventions that may be missing the mark and may be ineffective. Marital Suffocation Theory [10] posits that the meanings and expectations of marriage or committed relationships have changed over the last century to include meeting needs for intimacy and passion with autonomy and personal growth. Given this demand on relationships, some believe that this has destabilized the institution of marriage. As relationship science, family science, and couple and family therapy have “bought” into these ideas about relationships, our education programs and interventions may be expecting the same thing from relationships and trying to mold couples into what may be unrealistic forms. Certainly, if we continue to not be more careful in our assessments of relationships, we may be unwittingly fostering these myths.

### 1.1. Relationship Satisfaction/Quality/Adjustment in Older Couples

Understanding the specifics of relationship adjustment or quality among older adult couples is a second major gap in the literature. Again, most of the literature would suggest that satisfying relationships in later life promote health (i.e., 4, 6, 8). However, what a high-quality relationship might look like in later adulthood is lacking in the literature. Buhler, Kraus, and Orth [11] conducted a meta-analysis of relationship satisfaction across the lifespan and concluded that when considering chronological age, relationship satisfaction decreased from 20 to 40 then increased until about 65, where it plateaued. When considering relationship length, relationship satisfaction showed a decrease by year 10, then increased by year 20, and then decreased again. The most significant moderators were the presence of children and the assessment of satisfaction used. When using a global (unidimensional) measure of satisfaction rather than a multidimensional measure, satisfaction was higher at the outset, but the measurement type had no effect on the change in satisfaction over time.

Kiecolt-Glaser [5] suggested that older couples feel closer to their partner and feel more satisfied in their relationship than younger couples. It appears that older couples’ positive affect is less disrupted by disagreements than in younger couples, and they rate their partner’s behavior as more positive than independent raters. Braun et al. [12] expected that perceived reciprocity would decrease within older married couples because of normal declines due to aging. They found very little difference in perceived reciprocity in older vs. younger couples and no difference in relationship satisfaction. Researchers have concluded that the unique contributions of older age and relationship duration remain unclear [6]. That is, because “gray divorce” has been increasing [13], it has not been determined if it is the length of a relationship that might lead to the characteristics described or the age of the participants.

Other authors have used more nuanced definitions of relationship adjustment or satisfaction that have included both positive and negative dimensions. Otero et al. [3] suggested that positive interchanges have been shown to buffer against the adverse effects of negative affects. These authors coded behaviors among older couples that represented positivity resonance, which included shared positive affect, mutual care and concern, as well as behavioral and biological synchrony. Using this coding scheme, Otero et al. [3] found that positive resonance present during a conflict conversation was associated with relationship satisfaction even after controlling for positive and negative affect present during the conflict conversation. Cazzell et al. [14], using a large representative sample of couples (not just older couples), found that perceived negative interactions were not directly associated with relationship satisfaction but that the interaction of perceived positive and negative interactions was associated with relationship satisfaction. Perceived negative interactions were associated with decreased relationship satisfaction only when perceived positive interactions were low. Cazzell et al. [14] showed evidence for this effect for both actors (i.e., husband’s perceived negative and positive interactions) as well as a partner effect (wife’s perceived positive interaction with husband’s perceived negative interactions). The Cazzell et al. [14] study is rare in that it included dyadic data; what Carr and Utz [15] suggest will enhance our understanding of how relationships affect older adult well-being.

To summarize, as a field, we continue to debate what constitutes a satisfying relationship, and we know even less about what constitutes a satisfying relationship among older adults. A promising direction in research has been to investigate how positive and negative characteristics in relationships interact rather than simply adding these together to determine a score on a relationship satisfaction measure. If we have a better understanding of how positive and negative characteristics interact in relationships, and more specifically in relationships for older adults, we can provide better education and interventions when relationships are less satisfying, especially given that relationship satisfaction is associated with physical and mental health for older adults.

### 1.2. Purpose

Thus, the purpose of the current study was to explore the associations among positive and negative relationship indicators and their interactions with both relational and individual outcomes in a sample of older adult couples utilizing the NSHAP data [16]. We hypothesized that more negative characteristics would have a negative impact on relationship happiness, psychological symptoms, and physical health only in the presence of fewer positive characteristics in older adult couples.

## 2. Materials and Methods

### 2.1. Sample

For this project, we used the National Social Life, Health and Aging Project (NSHAP) dataset at Wave 2 [16]. NSHAP is a longitudinal study of health and social factors that influence the well-being of older Americans including physical health and illness, medication use, cognitive function, emotional health, sensory function, health behaviors, social connectedness, sexuality, and relationship quality. A nationally representative sample of adults aged 57 to 85 was interviewed in 2005 and 2006 for Wave 1 of the study. These same adults along with their spouses or cohabiting partners and those who did not respond at Wave 1 were interviewed in 2010–2011 for Wave 2 of the study (n = 3400). The response rate for Wave 1 respondents was 89%, and the conversion rate for Wave 1 nonresponders was 26% [17]. Only the dyad data from Wave 2 was used for this project.

Using the strata, cluster, and couple weight to make the sample representative of the US, and accounting for nonresponse, the sample consisted of 955 different sex dyads. Males were 71.10 years of age on average (se = 0.405), and females were 67.58 years on average (se = 0.387). A total of 35% of males had a bachelor’s degree or more, with 26% having a vocational certificate or some college and 23% having a high school diploma. A total of 24% of females had a bachelor’s degree or more, with 39.5% having a vocational certificate or some college and 25% having a high school diploma. Of the couples, 95% were married, with 3.9% cohabiting or dating. With weights to make the sample representative of the population, 83% of the couples were non-Hispanic White, 6.5% were non-Hispanic Black, and 7.33% were Hispanic. The couples reported an average income of USD 84,534 (se = USD 8587.714). Given the distribution of income, we used a log transformation in the analyses.

### 2.2. Instruments

There were several items that represented both positive and negative relationship characteristics that are often used in relationship satisfaction/quality/adjustment surveys.

#### 2.2.1. Time Spent Together

This item asked “Some couples like to spend their free time doing things together, while others like to do different things in their free time. What about you and your current partner? Do you like to spend free time doing things together or doing things separately?” Options were: 1, together; 2, some together some different; and 3, different/separate things. We recoded this to −1 for different/separate things, 0 for some together some different, and 1 for together.

#### 2.2.2. Social Network Placement of Spouse/Partner

Participants were asked to list the people in their social network in order of priority. Once they completed their list, they were asked on which line their spouse/partner appeared with options 1–6.

#### 2.2.3. Positive Relationship Characteristics

Several items in the survey were related to positive relationship characteristics, including: (1) “How often can you open up to your partner if you need to talk about your worries? Would you say never, hardly ever or rarely, some of the time or often?”; (2) “How often can you rely on your partner for help if you have a problem?”; (3) “How emotionally satisfying do you find your relationship with him/her?”; (4) “How often do you think that things be-tween you and your partner are going well?” All items used a Likert-type response ranging from 1 to 4. We took the average of the items, with higher scores indicating more positive characteristics. The internal consistency reliability of the items was 0.65 for wives, and it was lower for husbands at a value of 0.56. These total scores correlated with the global relationship happiness scores in the expected direction, where the values were 0.520 for wives and 0.37 for husbands. This provides some evidence for the validity of using these items as a scale score.

#### 2.2.4. Negative Relationship Characteristics

Several items in the survey seemed related to negative relationship characteristics, including: (1) How often does your partner make too many demands on you? (2) How often does your partner criticize you? (3) How often does your partner get on your nerves? Again, a Likert response was used, ranging from never to often. We used the average of the items with higher scores suggesting more negative characteristics. The internal consistency reliability of the items was 0.67 for wives and 0.66 for husbands. These scores were correlated with the global relationship happiness score in the expected direction, where the values were −0.347 for wives and −0.24 for husbands. Again, this provides some evidence for the use of these items as a scale score.

#### 2.2.5. Physical Health

The survey used self-rated physical health: “Would you say your health is excellent, very good, good, fair, or poor?” We recoded this to 1 for excellent and very good and 0 for good, fair, or poor.

#### 2.2.6. Mental Health

Depressive symptoms were assessed using the CESD 11 items [18,19]. The Chronbach alpha was 0.79 for wives and 0.779 for husbands in this sample. Anxiety symptoms were assessed with the Hospital Anxiety and Depression Scale [20], using only the 7 items that assessed anxiety. The Chronbach alpha was 0.73 for both wives and husbands. Stress was assessed with the Perceived Stress Scale (PSS) [21]. The Chronbach alpha was 0.57 for wives and 0.60 for husbands. Higher scores indicated more symptoms.

#### 2.2.7. Relationship Happiness

There was a single item to assess global relationship happiness, similar to the single-dimension measures used in previous studies to assess relationship satisfaction [1]. We used this item to show how the positive and negative characteristics interacted to be predictive of an overall assessment of relationship satisfaction. The item was “Taking all things together, how would you describe your relationship with current partner on a scale from 1 to 7 with 1 being very unhappy, and 7 being very happy.”

### 2.3. Data Analysis

We used an Actor Partner Interdependence Model with Moderation (APIMoM) [22]. This model can be seen in Figure 1. We tested the larger model and then simplified the model by testing for equality constraints in the actor and partner effects. The APIMoM tests both the actor (within person) effects and partner (between person) effects simultaneously. There are a number of different interaction terms that can be used: actor interaction and multiple partner interaction terms. For this project, we used only the actor interaction terms (i.e., female partner’s positive by negative relationship characteristics) given the sample size and more exploratory nature of the project.

The dyads were conceptually distinguishable, given that they identified as male and female. We also tested for empirical distinguishability using the ISAT model [23]. The fully saturated model with all means, variances, and covariances in the model free to vary between the partners was tested against a model that constrained the means, variances, and covariances equivalent between the partners. If the chi-square of the constrained model is significant, it suggests a loss of fit between the fully free model and the constrained model. In the current project, the χ^2^ difference was significant (Δχ^2^(54) = 136.56), suggesting that the dyads were empirically distinguishable based on gender. With distinguishable dyads, analyzing the APIMoM with structural equation modeling is appropriate.

## 3. Results

The means (standard error) of the variables of interest can be seen in Table 1 for the male and female partners. The correlations among the variables can be seen in Table 2 for both within partners (2a) and between partners (2b). In Table 2, the male partners’ correlations are above the diagonal and the female partners’ correlations are below the diagonal. The male partner’s relationship happiness was negatively associated with the negative relationship characteristics and was positively associated with the positive characteristics, which would be expected. It was also associated with time spent together and negatively associated with psychological symptoms. Negative relationship characteristics were negatively associated with time spent and positively associated with psychological symptoms. Positive characteristics were positively associated with time spent, negatively associated with psychological symptoms, and positively associated with physical health. Time spent and position in the social network were not associated with psychological symptoms. The psychological symptoms were intercorrelated and negatively associated with physical health. The pattern of correlations was the same for the female partners.

The correlations between the couple can be seen in Table 2. The correlations between the partner’s scores on all of the measures can be seen in the diagonal of the table; as can be seen, all variables were significantly and positively correlated between the partners. This suggests, for example, that the male partner’s perspective on negative characteristics in the relationship is associated with the female partner’s perspective on negative characteristics in the relationship. As one score increases, so does the other score. The female partner’s relationship happiness (column 1 in Table 2) was negatively associated with the male’s negative characteristics, positively associated with the male’s positive characteristics, positively associated with the male’s time spent, positively associated with where the male partner placed the spouse in their social network, negatively associated with the male partner’s depressive symptoms, anxiety and stress, and positively associated with the male partner’s health. The male partner’s relationship happiness (row 1 of Table 2) was negatively associated with the female partner’s negative characteristics, positively associated with the female partner’s positive characteristics and time spent, and negatively associated with the female partner’s depressive symptoms and anxiety. The remaining sets of correlations were in the expected direction, with some differences between how male partners’ scores were associated with female partners’ scores and vice versa. For example, male partners’ depressive symptoms were associated with female partners’ negative and positive characteristics, while female partners’ depressive symptoms were only negatively associated with male partners’ positive characteristics.

### 3.1. APIMoM Results

#### 3.1.1. Relationship Happiness

The associations among the explanatory variables and the outcomes varied. These can be seen in Table 3. First, we estimated a free to vary model with all paths free to be estimated. We then tested the equivalence of the actor effects (within person effects) using a chi-square difference test. If the chi-square difference was not significant, then the actor effects could be considered equivalent. We then tested the partner (between person effects) for equivalence in the same way. For male partner relationship happiness, his positive and negative relationship characteristics and their interaction were significantly predictive along with female partner’s negative relationship characteristics and his interpretation of time spent together. For female partner relationship happiness, the effects were equivalent with positive and negative characteristic actor effects and their interaction, along with male partner negative relationship characteristics and her interpretation of time spent together. The final relationship happiness model fit the data well (χ^2^(14) = 9.62; RMSEA = 0.00; CFI = 1.0; TLI = 1.0; SRMR = 0.014). The model explained about 19.5% of the male partner’s relationship happiness and 25.4% of the female partner’s relationship happiness. It should be noted that these positive and negative relationship characteristics, often used in multidimensional measures of relationship satisfaction or adjustment, explained only moderate amounts of variance in a more global measure of relationship happiness.

The moderating effect of positive relationship characteristics on the association between negative relationship characteristics and relationship happiness can be seen in Figure 2, showing a regions of significance plot. The solid line is the estimated line, while the two dashed lines represent the 95% confidence intervals. This plot is for the male partner, but as the estimates are equivalent, the plot would be the same for female partners. The positive characteristics are centered at the mean on the *x*-axis, so zero is the mean score. The *y*-axis is the adjusted effect or association between negative characteristics and relationship happiness. The association is zero or not significant when positive characteristics are about 0.5 standard deviations above the mean. That is, when partners rate positive characteristics at about half a standard deviation above the mean, there is no relationship between negative characteristics and relationship happiness. As positive characteristic ratings fall below the mean, then the association between negative characteristics and relationship happiness becomes negative as would be expected. What should be noted is that when positive characteristics are rated at one standard deviation above the mean, negative characteristics become positively associated with relationship happiness. It seems that for these couples to rate their relationship as happier, they not only need to perceive positive characteristics but negative characteristics also.

#### 3.1.2. Depressive Symptoms

In the model with depressive symptoms as the outcome, the actor effects were again equivalent with positive and negative relationship characteristics, but their interaction was not associated with depressive symptoms for both male and female partners. There was a partner interaction effect; however, for male partners, the female partner’s interaction term was significantly associated with the male’s depressive symptoms. Where the male partner placed his partner in his social network and income were significant predictors of male partner depressive symptoms as well. For female partners, the male’s education, their income, and where she placed her spouse in her social network were also significant predictors of female partner depressive symptoms. The final model for depressive symptoms fits the data well (χ^2^(7) = 3.70; RMSEA = 0.00; CFI = 1.0; TLI = 1.0; SRMR = 0.006). The model explained about 15% of the variance in male partner’s depressive symptoms and 12.7% of the variance in female partner’s symptoms.

The regions of significance plot for male partners’ depressive symptoms and female partners’ relationship characteristics can be seen in Figure 3. The *x*-axis is female partners’ positive relationship characteristics, and the *y*-axis is the association between male partners’ depressive symptoms and female partners’ negative relationship characteristics. The association between the female partner’s negative relationship characteristics and male partner’s depressive symptoms is at zero when female partners rate the positive relationship characteristics about 0.3 standard deviations above the mean. That is, there is no association between negative characteristics and male depressive symptoms when female partners rate positive characteristics more highly. The association between female partners’ negative characteristics and male partners’ depressive symptoms become significantly positive (as expected) only when the female partner rates positive characteristics about 0.5 standard deviations below the mean. Because the confidence intervals encompass zero toward the “right” side of the figure, there continues to be no association between negative characteristics and male depressive symptoms as female partners rate positive characteristics more highly. Thus, the interaction of positive and negative relationship characteristics are associated with male partner’s depressive symptoms. Positive and negative relationship characteristics are directly associated with both partners’ depressive symptoms as expected but not the interaction between them.

#### 3.1.3. Anxiety Symptoms

In the remaining three models (anxiety, stress, and physical health), the interaction terms were not significant. For the anxiety symptoms model (χ^2^(6) = 4.82; RMSEA = 0.00; CFI = 1.0; TLI = 1.0; SRMR = 0.006), negative relationship characteristics by the actor were positively associated with anxiety in the actor for both partners. These effects were equivalent between the partners. The model explained about 6.6% of both partners’ anxiety symptoms.

#### 3.1.4. Stress

For the stress model (χ^2^(14) = 12.84; RMSEA = 0.00; CFI = 1.0; TLI = 1.0; SRMR = 0.012), positive and negative relationship characteristics were associated with stress for both partners, and these actor effects were equivalent. Positive relationship characteristics were negatively associated with stress, while negative relationship characteristics were positively associated with stress. Higher levels of education were negatively associated with stress equivalently for both male and female partners. The model explained about 7% of the variance in both partners’ stress.

#### 3.1.5. Physical Health

The model for physical health (χ^2^(0) = 0; RMSEA = 0.00; CFI = 1.0; TLI = 1.0; SRMR = 0.00) was not equivalent between the male and female partners. The only significant predictors for male physical health were income and where the female partner placed the spouse in their social network. If the female partner did not place the spouse at a value of one, the male partner was less likely to report that they had good to excellent health. For the female partner, the male partner’s positive relationship characteristics, his education, their income, and her education were all significant predictors of the female partner’s physical health. The model explained about 17.5% of the variance in the male partner’s physical health and 30.3% of the female partner’s physical health. Thus, for these health outcomes (anxiety, stress, and physical health) the interaction of positive and negative relationship characteristics were not associated, while there were direct associations with positive and negative characteristics.

## 4. Discussion

The purpose of this study was to explore how the interaction of positive and negative relationship characteristics among older adult couples were associated with relationship and individual outcomes. We hypothesized that the interaction of positive and negative characteristics would be predictive of these outcomes. That is, relationship happiness would only be negatively associated with negative characteristics if positive characteristics were low as they have been found previously [14]. Our findings supported this.

We found that male partners’ depressive symptoms were predicted by the interaction of female partners’ perception of positive and negative relationship characteristics. When female partners rated positive characteristics more highly, the association between negative characteristics and male depressive symptoms was not significant. That is, only when female partner’s positive characteristics were low did her perspective on negative characteristics matter for male partner depressive symptoms. There is a fairly large amount of literature that suggests that relationship satisfaction and depressive symptoms are reciprocally associated with some evidence that wives’ depressive symptoms are more highly associated with husbands’ relationship satisfaction than vice versa [24]. Other research has suggested that when controlling for other relationship characteristics (i.e., differentiation of self), the association between depressive symptoms and relationship satisfaction is no longer significant for either partner [25]. The results of this project would suggest that in older couples, husband’s depressive symptoms appear to be more sensitive to the “balance” of positive and negative characteristics of the relationship, at least from their wives’ perspectives. Wives’ depressive symptoms were associated with their own perceptions of positive and negative relationship qualities but not the interaction of the two. Thus, for wives, their depressive symptoms were higher when negative qualities were higher and were lower when positive qualities were higher.

The interaction of positive and negative characteristics was not predictive of anxiety symptoms, stress, or physical health, but there were direct effects of positive and negative characteristics on these outcomes for the most part. Only the ratings of the negative characteristics were associated with anxiety symptoms. Both positive and negative characteristics were associated with stress. Neither positive nor negative characteristics of the relationship were associated with self-rated physical health for male partners, but male partners’ perception of positive characteristics were significantly associated with female partners’ physical health. Of note for male partners, where their partner placed them in their social network was associated with male partners’ health. Previous research has suggested that relationship satisfaction or a quality relationship is supportive of better physical health [5]. In this project, this appears to be the case for female partners but less so for male partners. It may be that when male partners have health problems, female partners rate them less highly on their social network. The direction of these relationships cannot be determined with cross-sectional data.

The overall purpose of this project was to assess relationship satisfaction/adjustment/quality in a more nuanced way by using both positive and negative relationship characteristics rather than a global measure of satisfaction or a measure that determined what a satisfying, adjusted, or quality relationship includes. The Dyadic Adjustment Scale, one of the most often-used scales if researchers are using scales for relationship quality [1], asks about certain aspects of the relationship (i.e., frequency of spending time together, frequency of conflicts, topics of conflicts, etc.). The scores from these items are then summed to create an overall score of adjustment. The results of the current project would suggest that this strategy may not capture “reality” for most couples. These data would suggest that at least older couples have both positive and negative characteristics in their relationships; although how much time they spend together was associated with these, time spent together was not predictive of any outcomes except for the global relationship happiness. It is also true that not all couples placed each other as first on their list of people in their social network, and this item was not highly associated with other relationship characteristics.

The positive and negative characteristics interacted such that in the presence of more positive characteristics, negative characteristics were not associated with a global measure of relationship satisfaction. Cazzell et al. [14] demonstrated similar findings in a large sample of couples (including older and younger couples). Of note, however, is that these positive and negative characteristics and their interaction as well as the other variables in the model explained about 20–25% of the variance in global relationship happiness. These older couples use something else to determine their level of overall happiness in their relationships than what many of the multidimensional measures of relationship adjustment use as indicators of relationship satisfaction, quality, or adjustment. What that something else is remains to be understood.

Braun et al. [12], using social exchange equity theory, expected that there would be a decrease in the perception of reciprocity in older adult couples simply due to limitations of aging. They did not find this to be the case. Thus, the theories we use to explain relationship satisfaction appear to have limitations when it comes to their application to older couples. Given our lack of theoretical conceptualizations around relationship satisfaction, quality, or adjustment more generally, it will be important to develop theory to better understand how relationships develop over time and what constitutes high-quality or well-adjusted relationships in older adult couples.

As with any secondary data analysis research, several limitations exist. We used items from the original study’s methodology that best reflected relationship quality characteristics. The cross-sectional nature of the analysis does not allow us to infer causality. The reliabilities of some of the scales seemed low, which also brings limitations to the findings. Partly, these scales were short and the sample was large, which biases the results to some degree. That is, with a larger sample size, there is more statistical power to detect any discrepancies in the inter-item correlations, which will affect the internal consistency reliability. Using longitudinal designs to investigate the causal associations and temporal direction of the constructs used in this study would benefit future research. Also, using instruments that represent more positive and more negative characteristics in relationships may also be an avenue for future research.

Nevertheless, these findings have implications for relationship researchers, educators, and clinicians. Similar to ideas of Gottman [26], there appears to be a ratio of positive and negative characteristics that support overall relationship satisfaction as well as mental health, at least for males. What this ratio is exactly was not determined in this project, but future research could investigate the “tolerance” of negative and positive characteristics for relationship satisfaction. That is, future research could determine at what point the balance tips so that negative characteristics begin to decrease relationship satisfaction overall or when there is not enough positive to outweigh the negative. If we have information about this ratio of positive to negative, then teaching couples about this ratio may be a useful way for relationship educators to proceed. Rather than trying to have every couple relationship look a particular way, the education would be tailored and would focus on processes in the relationship that are positive and processes that are negative and how to balance them. Clinicians could also focus on balancing positive and negative characteristics in relationships rather than solely aiming to decrease conflict. By addressing the balance of characteristics, clinicians could help older couples enhance the satisfaction of their relationship and therefore improve health outcomes.

More importantly, the results of the current study point to the need to continue to refine our definitions and assessments of relationship satisfaction, adjustment, or quality. The results of this study showed that items often used in relationship adjustment assessments only explained moderate amounts of variance in a global satisfaction measure. Without clearer evidence that there are specific ingredients in relationships that could explain overall satisfaction, it is a mistake to use assessments that focus on these ingredients rather than on the perception of relationship satisfaction among couple members.

## Figures and Tables

**Figure 1 behavsci-14-01017-f001:**
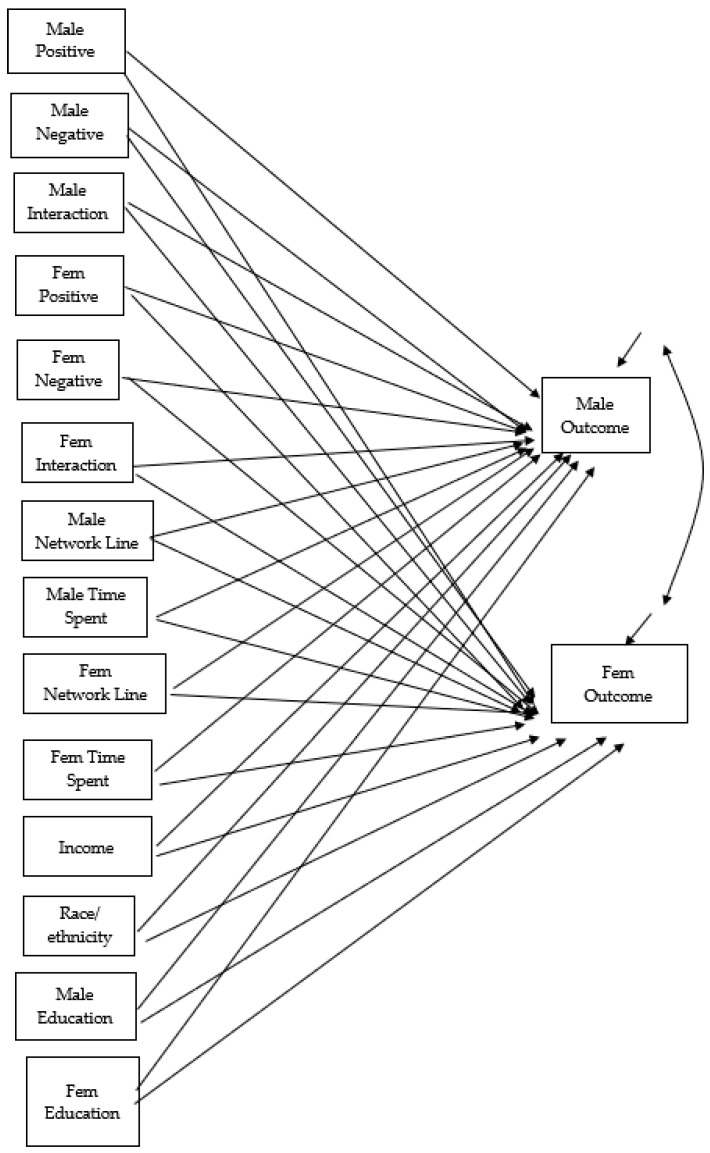
APIMoM model for the outcomes.

**Figure 2 behavsci-14-01017-f002:**
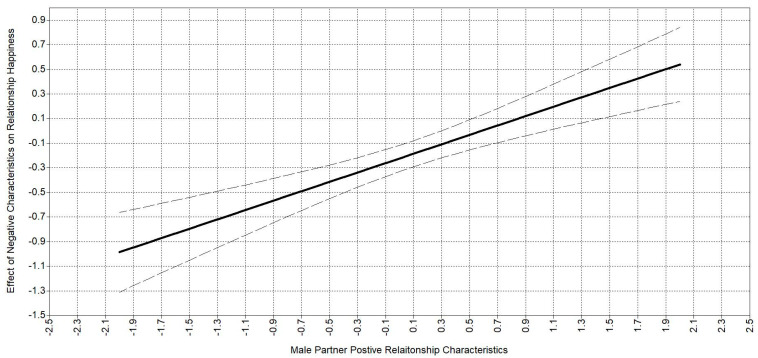
Regions of significance plot for positive characteristics and relationship happiness moderated by negative characteristics.

**Figure 3 behavsci-14-01017-f003:**
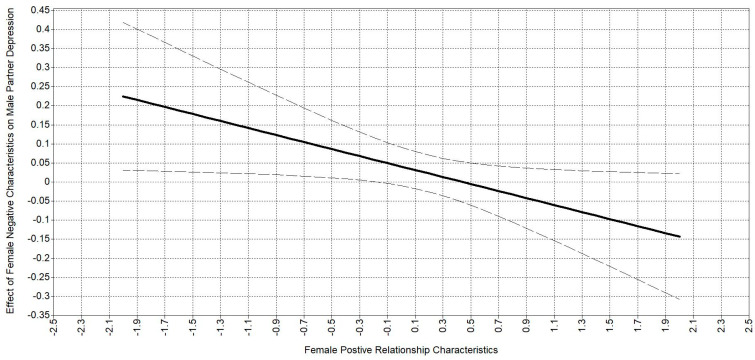
Regions of significance plot for female positive characteristics and male depressive symptoms moderated by female negative characteristics.

**Table 1 behavsci-14-01017-t001:** Weighted means (se) of the variables of interest in the study for male and female partners.

	Male Partner	Female Partner
Positive Relationship	2.94 (0.016)	2.82 (0.018)
Negative Relationship	1.33 (0.024)	1.22 (0.026)
Network Placement Spouse	1.16 (0.058)	1.12 (0.046)
Relationship Happy	6.46 (0.034)	6.27 (0.051)
Depressive Symptoms	1.37 (0.013)	1.45 (0.017)
Anxiety	1.65 (0.022)	1.73 (0.025)
Stress	1.77 (0.026)	1.82 (0.034)
Time Spent	44.25% some together some separate/47% together	43.8% some together some separate/44.27% together
Physical Health	75.3% very good to excellent	78.85% very good to excellent

**Table 2 behavsci-14-01017-t002:** (a). Correlations among the study variables. Males above the diagonal; females below the diagonal. (b) Correlations of study variables between the partners.

(a)									
	1	2	3	4	5	6	7	8	9
1. Rl Happy		−0.232 ***	0.351 ***	0.154 ***	0.044	−0.124 ***	−0.119 ***	−0.091 ***	0.029
2 Negative Characteristics	−0.335 ***		−0.343 ***	−0.180 ***	−0.034	0.250 ***	0.243 ***	0.162 ***	−0.041
3. Positive Characteristics	0.476 ***	−0.440 ***		0.248 ***	0.060	−0.186 ***	−0.125 ***	−0.149 ***	0.107 ***
4. Time Spent	0.254 ***	−0.236 ***	0.343 ***		−0.036	−0.055	−0.029	0.017	0.012
5. Position in Social Network	−0.021	0.040	−0.019	−0.020		0.013	−0.026	−0.034	0.012
6. Depressive Symptoms	−0.161 ***	0.200 ***	−0.257 ***	−0.059	0.032		0.416 ***	0.344 ***	−0.276 ***
7. Anxiety	−0.119 ***	0.187 ***	−0.177 ***	−0.018	−0.005	0.422 ***		0.495 ***	−0.111 ***
8. Stress	−0.131 ***	0.118 ***	−0.148 ***	−0.050	−0.072 *	0.317 ***	0.517 ***		−0.183 ***
9. Physical Health	0.029	−0.044	0.162 ***	0.027	−0.005	−0.303 ***	−0.141 ***	−0.192 ***	
**(b)**									
	**1F**	**2F**	**3F**	**4F**	**5F**	**6F**	**7F**	**8F**	**9F**
1. MRl Happy	0.155 ***	−0.187 ***	0.139 ***	0.112 ***	0.015	−0.077 *	−0.079 *	−0.044	0.024
2 MNegative Characteristics	−0.175 ***	0.279 ***	−0.192 ***	−0.114 ***	0.009	0.037	0.079 *	0.033	−0.007
3. MPositive Characteristics	0.148 ***	−0.211 ***	0.306 ***	0.132 ***	0.027	−0.128 ***	−0.065	−0.021	0.184 ***
4. MTime Spent	0.126 ***	−0.153 ***	0.147 ***	0.411 ***	−0.036	−0.031	0.021	0.002	0.008
5. MPosition in Social Network	0.071 *	−0.056	0.050	−0.017	0.118 ***	−0.005	−0.041	−0.022	−0.008
6. MDepressive Symptoms	−0.171 ***	0.147 ***	−0.155 ***	−0.097 **	0.012	0.172 ***	0.105 ***	0.141 ***	−0.087 **
7. MAnxiety	−0.130 ***	0.090 **	−0.125 ***	−0.037	−0.024	0.148 ***	0.223 ***	0.200 ***	−0.042
8. MStress	−0.111 ***	0.085 *	−0.152 ***	−0.033	0.000	0.148 ***	0.171 ***	0.233 ***	−0.105 **
9. MPhysical Health	0.151 ***	−0.139 ***	0.141 ***	0.084 *	−0.034	−0.109 ***	−0.020	−0.093 **	0.150 ***

M = male partner; F = female partner; * *p* < 0.05; ** *p* < 0.01; *** *p* < 0.001.

**Table 3 behavsci-14-01017-t003:** Path estimates (standard errors) for the APIMoM models. * *p* < 0.05.

	Relationship Happiness		Depression		Anxiety		Stress		Physical Health	
Path	Male	Female	Male	Female	Male	Female	Male	Female	Male	Female
Male Positive	0.818 (0.088) *	−0.083 (0.062	−0.106 (0.034) *	−0.057 (0.048)	−0.086 (0.057)	−0.019 (0.047)	−0.136 (0.057) *	0.001 (0.047)	0.212 (0.138)	0.317 (0.121) *
Male Negative	−0.223 (0.055) *	−0.163 (0.062) *	0.117 (0.022) *	−0.023 (0.022)	0.149 (0.028) *	−0.007 (0.027)	0.132 (0.028) *	−0.011 (0.028)	0.168 (0.109)	−0.024 (0.084)
Male Interaction	0.382 (0.075) *	0.144 (0.120)	−0.085 (0.054)	0.083 (0.050)	−0.029 (0.044)	−0.025 (0.049)	−0.004 (0.054)	−0.077 (0.068)	−0.002 (0.139)	−0.039 (0.158)
Fem Postive	−0.083 (0.062)	0.818 (0.088) *	0.009 (0.030)	−0.106 (0.034) *	−0.019 (0.047)	−0.086 (0.057)	0.001 (0.047)	−0.136 (0.057) *	0.091 (0.134)	0.189 (0.123)
Fem Negative	−0.163 (0.062) *	−0.223 (0.055) *	0.041 (0.026)	0.117 (0.022) *	−0.007 (0.027)	0.149 (0.028) *	−0.011 (0.028)	0.132 (0.028) *	−0.218 (0.113)	−0.075 (0.089)
Fem Interaction	0.144 (0.120)	0.382 (0.075) *	−0.092 (0.044) *	0.008 (0.058)	−0.025 (0.049)	−0.029 (0.044)	−0.077 (0.068)	−0.004 (0.054)	−0.179 (0.118)	0.035 (0.117)
Male Network Placement	−0.021 (0.019)	0.023 (0.020)	0.022 (0.009) *	0.004 (0.013)	−0.013 (0.022)	−0.017 (0.014)	−0.014 (0.018)	−0.004 (0.017)	−0.015 (0.049)	−0.042 (0.034)
Male Time Spent	0.116 (0.051) *	−0.003 (0.044)	0.008 (0.017)	0.021 (0.021)	0.016 (0.029)	0.062 (0.034)	0.023 (0.026)	0.002 (0.028)	−0.022 (0.077)	−0.040 (0.084)
Male Ed	0.020 (0.028)	−0.062 (0.036)	−0.021 (0.013)	−0.043 (0.014) *	−0.045 (0.027)	−0.002 (0.028)	−0.098 (0.022) *	−0.030 (0.081)	0.027 (0.097)	0.138 (0.055) *
Income	0.013 (0.053)	0.013 (0.053)	−0.045 (0.014) *	−0.045 (0.014) *	−0.003 (0.025)	−0.008 (0.035)	−0.008 (0.028)	−0.008 (0.028)	0.270 (0.084) *	0.318 (0.057) *
Race/Ethnicity	−0.180 (0.148)	−0.180 (0.148)	0.018 (0.050)	0.018 (0.050)	0.028 (0.080)	−0.006 (0.079)	0.038 (0.081)	0.038 (0.081)	−0.067 (0.133)	−0.048 (0.157)
Fem Network Placement	0.023 (0.020)	−0.021 (0.020)	0.011 (0.012)	0.022 (0.009) *	0.002 (0.015)	0.003 (0.016)	−0.004 (0.017)	−0.014 (0.018)	−0.059 (0.030) *	−0.042 (0.034)
Fem Time Spent	−0.003 (0.044)	0.116 (0.051) *	−0.043 (0.024)	0.008 (0.017)	0.005 (0.028)	0.039 (0.037)	0.002 (0.028	0.023 (0.026)	0.119 (0.077)	0.003 (0.082)
Female Ed	−0.062 (0.036)	0.020 (0.028)	0.007 (0.015)	−0.021 (0.013)	−0.012 (0.030)	−0.033 (0.028)	−0.030 (0.018)	−0.098 (0.022) *	0.134 (0.073)	0.148 (0.056) *

## Data Availability

This is a secondary data analysis study. The National Social Life and Healthy Aging Project (NSHAP) data are deidentified for public use, and restricted data are available for use upon approval by the Inter-university Consortium for Political and Social Research (ICPSR). The produced data on this study’s outcomes will be submitted to the ICPSR. Data sharing will be in compliance with the Data Materials and Distribution Agreement with ICPSR.

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
