# Peer review of "The Interaction of Positive and Negative Relationship Characteristics and Their Association with Relationship and Individual Health Outcomes in Older Couples"

_behavsci, 2024, doi:10.3390/bs14111017_

Round 1

Reviewer 1 Report

Comments and Suggestions for Authors

The article addresses a relevant topic and contributes to the study of relationship quality in mature couples, through a theoretical and methodological reflection, followed by a well-structured analysis. The study employs a quantitative approach, with rigorous statistical analyses based on secondary data.

In my opinion, there are a few areas that could benefit from improvement. The results are presented in a highly technical manner, which might hinder comprehension for readers not deeply familiar with the methods of analysis. Including a summary table or a concise presentation of key findings, especially in the conclusions, could enhance readability and the overall impact of the study.

Additionally, the initial section addressing the debate regarding the concept of "marital quality" could be better integrated with the rest of the paper. This section tends to lose its connection as the analysis progresses, and  a stronger link to the conclusions would improve coherence.

The article acknowledges some limitations, as mentioned in the conclusions. Nevertheless, it would be beneficial to elaborate further on these limitations, providing more in-depth insights into their implications for the study’s results and potential future research directions.

As minor revisions, I suggest reconsidering the use of the term “health outcome” in the title of the article to ensure it accurately reflects the content of the study. Additionally, Table 3 requires better formatting to improve its readability and presentation.

Author Response

Comment: In my opinion, there are a few areas that could benefit from improvement. The results are presented in a highly technical manner, which might hinder comprehension for readers not deeply familiar with the methods of analysis. Including a summary table or a concise presentation of key findings, especially in the conclusions, could enhance readability and the overall impact of the study.

Response:  A summar of the results been added and simplified language has been added as well.

Comment: Additionally, the initial section addressing the debate regarding the concept of "marital quality" could be better integrated with the rest of the paper. This section tends to lose its connection as the analysis progresses, and  a stronger link to the conclusions would improve coherence.

Response: This link has been emphasized more clearly in the results section and in the discussion.

Comment:  The article acknowledges some limitations, as mentioned in the conclusions. Nevertheless, it would be beneficial to elaborate further on these limitations, providing more in-depth insights into their implications for the study’s results and potential future research directions.

Response:  We have ncluded these.

Comment:  As minor revisions, I suggest reconsidering the use of the term “health outcome” in the title of the article to ensure it accurately reflects the content of the study. Additionally, Table 3 requires better formatting to improve its readability and presentation.

Response:  Table 3 has been revised.  We use the term "health outcomes" in the title to reflect both mental and physcial health, since it is already a long title we did not want to us that phrase. 

Reviewer 2 Report

Comments and Suggestions for Authors

See PDF
